# The Multifunctionality of Exosomes; from the Garbage Bin of the Cell to a Next Generation Gene and Cellular Therapy

**DOI:** 10.3390/genes12020173

**Published:** 2021-01-27

**Authors:** Surya Shrivastava, Kevin V. Morris

**Affiliations:** 1Center for Gene Therapy, City of Hope-Beckman Research Institute, Duarte, CA 91010, USA; sshrivastava@coh.org; 2Hematological Malignancy and Stem Cell Transplantation Institute at the City of Hope, Duarte, CA 91010, USA; 3School of Medical Science, Gold Coast Campus, Griffith University, Southport 4222, Australia

**Keywords:** exosome, therapeutics, nucleic acid payloads, nanoparticle

## Abstract

Exosomes are packaged with a variety of cellular cargo including RNA, DNA, lipids and proteins. For several decades now there has been ongoing debate as to what extent exosomes are the garbage bin of the cell or if these entities function as a distributer of cellular cargo which acts in a meaningful mechanistic way on target cells. Are the contents of exosomes unwanted excess cellular produce or are they selective nucleic acid packaged nanoparticles used to communicate in a paracrine fashion? Overexpressed RNAs and fragments of DNA have been shown to collect into exosomes which are jettisoned from cells in response to particular stimuli to maintain homeostasis suggesting exosomes are functional trash bins of the cell. Other studies however have deciphered selective packaging of particular nucleic acids into exosomes. Nucleic acids packaged into exosomes are increasingly reported to exert transcriptional control on recipient cells, supporting the notion that exosomes may provide a role in signaling and intracellular communication. We survey the literature and conclude that exosomes are multifunctional entities, with a plethora of roles that can each be taken advantage to functionally modulate cells. We also note that the potential utility of developing exosomes as a next generation genetic therapy may in future transform cellular therapies. We also depict three models of methodologies which can be adopted by researchers intending to package nucleic acid in exosomes for developing gene and cell therapy.

## 1. Introduction

Extracellular vesicles are biological materials released by cells, surrounded by a lipid bilayer membrane which lack a functional nucleus and vary in size range from 30 to 10,000 nm [1,2]. Based on their size, extracellular vesicles (EV) are classified into exomeres (30–50 nm) ([3,4]), exosomes (50 to 150 nm), micro vesicles (150 to 1000 nm), oncosomes (1000–10,000 nm) [5] and apoptotic bodies (100–5000 nm). In addition to size, these different categories also vary in their mechanism of production from cells and their molecular composition [6,7,8]. They can be predicted to vary in terms of range of their action and their half-life, although no study to our knowledge to date has done such a comparison. Out of the various EV subpopulations, exosomes are by far the most studied in terms of composition and adoption as a vehicle for delivery of biomolecules. Exosomes like other extracellular vesicles are composed of proteins, lipids and nucleic acid [9,10]. Biogenesis of exosomes (discussed extensively in recent review [11]) is aided primarily by ESCRT pathway proteins and, also, by ESCRT independent pathways. Although cues of initiation are not clear, cell membrane invaginates to form endosomes. Further inward budding of membrane of endosome gives rise to multivesicular bodies (MVB) or late endosomes. Late endosome is the stage of major cargo sorting and a platform for researchers to fortify exosomes with therapeutic cargo. Aided by cytoskeletal proteins, SNARE complexes and scaffolding proteins, MVB are transported to plasma membrane where MVB fuses with cell membrane to release exosomes out of cell [11]. Nucleic acids, i.e., messenger RNA (mRNA), micro RNAs (miRNAs) and long non-coding RNAs (lncRNAs) are packaged into exosomes and provide an extraordinary opportunity to disseminate protein coding mRNA and/or control gene expression (miRNA and lncRNA) in distal cells. We review and contrast the small RNA (miRNA, small nucleolar RNA (snoRNA), PIWI interacting RNAs (piRNA), tRNA, yRNA), circular RNA (circRNA), lncRNA, mRNA and DNA composition of exosomes along with their sorting mechanisms providing insights into the various pathways with regards to developing next generation gene and cell therapies. 

## 2. miRNA

Exosome associated miRNAs have been extensively profiled from virtually all possible sources including plasma [12], cerebrospinal fluid [13], milk [14], semen [15], urine [16], amniotic fluid [17] and bronchoalveolar lavage [18]. The miRNA composition in plasma derived exosomes is highly sensitive to change in microenvironment like exposure to gamma rays [19], cigarette smoke [20] and circadian rhythm [21]. Differentiation into specific cell lineages is also influenced by exosome miRNA profiles. For instance, exosomes derived from B cells, T cells and dendritic immune cells are comprised of miRNA populations that vary from those of their parent cells [22]. In another study, exosomes from the late stage of osteogenic differentiation of bone marrow derived MSC had a different miRNA expression profile in comparison with early stage MSC. These differentially expressed exosomal miRNAs were shown to regulate pathways involved in osteogenic differentiation [23]. Furthermore, virtually every stage of cancer progression starting from early signs of transformation to metastasis influences the miRNA profile of exosomes (reviewed in [24]. Owing to sensitivity of changing environment, exosome miRNA signatures have been shown to be biomarkers for various metabolic conditions like atrial fibrillation [25], renal graft function [26], pancreatic lesions [27], liver disease [28] and various types of cancers as reviewed in [29]. In addition to a prognostic tool, miRNAs have been shown to influence both local and distal gene regulation when packaged and delivered to cells via exosomes. Platelet derived exosomes containing miR-223, miR-339 and miR-21 can locally influence gene expression of platelet derived growth factor receptor β (PDGFRβ) in smooth muscle cells within blood vessels and reduce their proliferation to prevent potential stenosis in an atherothrombosis murine model [30]. While in another study, the heart-brain axis was found to be influenced by depletion of exosome bound miR-126 in endothelial cells of cerebral artery leading to increased cardiac dysfunction distally in murine model of stroke [31]. Other examples of functional exosome packaged miRNAs includes exosomal miR-21 mediated regulation of human PTEN [32] and transfer of the osteoclast-derived exosomal miR-214-3p to osteoblasts to inhibit bone formation [33].

There is now a large body of studies implicating exosome bound miRNAs effective in gene modulation which are distinct functions, independent of the excess production of particular miRNAs in the parent cell source. Because of these observations, exosomes are increasingly becoming vehicles of choice to deliver therapeutic miRNAs. Initially, investigators collected exosomes from cells which naturally showed higher miRNA packaging than others and demonstrated delivering of the particular miRNAs into target cells. Embryonic stem cell (ESC) derived exosomes were found to deliver miR-294 to the heart resulting in significant modulation of survival, proliferation and cardiac commitment of cardiac progenitor cells [34]. At the same time, enhanced cardiomyocyte survival and proliferation take place because of ESC exosome delivery that ultimately leads significant augmentation of cardiac regeneration in the heart following myocardial infarction [34]. Exosomes from placenta derived mesenchymal stem cells (PL-MSC) have also been shown to be of therapeutic value to Duchenne muscular dystrophy patients and mdx mice models whereby exosomes from PL-MSC rich in miR-29c induced differentiation of myoblast and inhibited fibrosis and inflammation [35]. 

While the studies to date suggest that exosomes can be used to deliver particular miRNA signatures to target cells, it became apparent that an ability to program particular miRNAs into exosomes would prove an important step in developing exosomes as therapeutics capable of modulating gene expression. Towards this goal, studies into the natural enrichment of miR-26c into exosomes led to the observation that miR-26c encapsulated exosomes were directed to skeletal muscle cells and heart via a muscle specific peptide [36]. This targeted delivery of exosomal miR-26c prevented muscle wasting and attenuated cardiomyopathy in mice model of chronic kidney disease [36]. Similar observations were noted with HEK293T derived exosomes which were enriched with miR-26a by electroporation. These miR-26a packaged exosomes were found to preferentially target liver cancer HepG2 cells via surface expression of Apo.A1 on exosomes [37]. Targeted delivery of miR-26a decreased HepG2 cell migration and proliferation [37]. However, simple over-expression of miRNAs does not always result in the sorting of particular miRNAs into exosomes. A recent study established a miRNA embedded motif AAUGC as a key sequence required to interact with RNA binding protein FMR1 and selectively load miRNAs into exosomes following intracellular inflammasome activation [38]. This miRNA signature and various other miRNA sequence motifs which have been observed to interact with exosomal trafficking proteins are described in detail (reviewed in [39]). Collectively, these studies all indicate that exosomal miRNA content can serve as a diagnostic as well as a therapeutic tool.

## 3. snoRNA

Small nucleolar RNA have been shown to impart post-transcriptional specific chemical modification like 2′-O-methylation or acetylation on complimentary RNA [40]. In particular 2′-O-methylation was shown to be important in determination of self RNA from non self (viral RNA) and prevention of RNA hydrolysis [41]. A vast majority of snoRNA ranging from 60 to 300 nucleotides in length are found in nucleus but recent sequencing studies have also identified their presence in exosomes. SnoRNAs have been found to make up ~0.07% of total RNA in the neuronal cell derived exosomes [42]. The composition of snoRNAs in exosomes was found to change and increase in exosomes in the presence of West Nile Infection, and was suggested to be a means to spread innate anti-viral effects throughout a local microenvironment [43]. Exosome bound snoRNAs also have the potential to become biomarkers for pancreatic cancer [44]. Specific enrichment of particular snoRNAs have been shown in CD47+ cells derived exosomes, however, no mechanism of snoRNA sorting has yet been deduced [45]. In a fascinating recently published murine model of parabiosis, exosomal Rpl13a snoRNA secreted from one of the parabiont had 2-O-methylation effect on rRNA in the other parabiont [46]. This peculiar study demonstrated that snoRNA packaging into exosomes is not a mere dump of cellular biproducts but rather is capable of affecting distal gene regulation.

## 4. PIWI Interacting RNA

PIWI interacting RNAs interact with PIWI domain containing proteins which can cleave RNA complimentary or silence transposable elements [47]. piRNA ranging from 21 to 35 nucleotides in length were first detected in exosomes by next generation sequencing from saliva in addition to snoRNA and miRNA and accounted for nearly 5% of total sequences mapped to known RNA [48] and about 1.31% in human plasma derived exosomes [49]. A variety of piRNAs were differentially expressed in exosomes of poly cystic ovary syndrome (PCOS) relative to control from human follicular fluid, suggesting these particular piRNAs could serve as prognostic biomarker for this condition [50]. Likewise, piRNA populations were also altered in plasma exosomes of cholangiocarcinoma and gall bladder carcinoma. Notably, piR-10506469 was found to be substantially increased in cholangiocarcinoma and gall bladder carcinoma patients, and was observed to decline following surgery to remove the cancerous mass, suggesting that piR10506469 may serve as a biomarker of disease progression [51]. Another piRNA, piR-004800, was found to be in excess in exosomes from bone marrow of multiple myeloma (MM) patients. This exosome bound piR-004800 has been observed to play an oncogenic role by influencing sphingosine-1 phosphate receptor signaling and promoting MM cell proliferation [52]. The observations with piRNAs packaged into exosomes suggest that these RNAs also exert a function on the exosome targeted cells.

## 5. tRNA

Fragments derived from tRNA (tRF) can act as translation inhibitors and were mechanistically shown to act by disrupting translation initiation complex [53]. Presence of tRFs have been detected in exosomes isolated from semen, urine [15,54], dendritic cells [55] and amounted to ~50% of total EV associated RNAs from HEK cells [56]. Plasma exosomes derived from osteoporosis patients have also been found to contain enriched levels of tRF-18, tRF-25 and tRF-38, providing a convenient diagnostic marker for osteoporosis [57]. Plasma exosomes isolated from liver cancer patients were found to contain elevated levels of fragments of tRNA-ValTAC-5, tRNA-ValAAC-5, tRNA-GlyTCC-5 and tRNA-GluCTC-5, and were also proposed to be good candidates for early diagnosis [58]. In an interesting study T cells were shown to expel tRFs into their exosomes which if retained in the cell can potentially lead to a blockage of T cell activation [59]. As retaining tRFs could be hazardous, researchers tried to understand tRF sorting and found that MCF-7 cells overexpressing tRFs from tRNA-Gly were enriched in exosomes [60]. This study proved that cellular overproduced tRFs were being actively transported into exosomes and shed from the cell. Utilizing this strategy, tRNA derived small non-coding RNA-10277 i.e., tsRNA-10277 was loaded into BMSC derived exosomes. These tsRNA-10277 enriched exosomes were shown to enhance osteogenic differentiation of recipient BMSCs [61]. Overall, from the available literature it appears that tRFs are a biproduct of cellular metabolism, expelled into exosomes, which have distal gene regulatory effect. 

## 6. Y-RNA

Y-RNA are small non-coding RNAs, which are roughly 100 nucleotides in length and shown to interact with chromosomal DNA in nucleus and aid in replication [62]. In cytoplasm, Y-RNA has been shown to aid in regulation of misfolded RNA by interacting with protein Ro [63]. Y-RNA fragments have been shown to be present in serum as specific fragments of 27 to 33 nucleotides in size derived from cleavage at internal loop of Y-RNA [64]. Y-RNA derived fragments were found to be enriched by ~75% in exosomes compared to source endothelial producer cells [65] and in malignant melanoma [66]. Exosomal RNY4 (one of the full-length Y-RNA) was proposed as a promising biomarker of anaplastic large cell lymphoma after testing 20 pediatric patient’s plasma derived exosomes [67]. Y-RNA hY4 was also found to be a highly abundant RNA species present in plasma exosomes from chronic leukocytic lymphoma (CLL) patients and that hY4 enriched exosomes skew normal monocytes towards CLL associated phenotypes due to the exosomal hY4 RNAs acting as a driver of TLR7 signaling [68]. Based on this study, another study proposed hY4 as link between malaria and Kaposi’s sarcoma herpesvirus (KSHV). In the malaria endemic regions, Y-RNA was elevated in exosomes from erythrocyte exposed to malarial parasites. These Y-RNA enriched exosomes can activate TLR7/8 pathway and induce latent KSHV [69]. A fragment of Y-RNA (EV-YF1) conferred cardio-protection when present in exosomes derived from cardio-sphere derived cells, by inducing IL10 expression in rat model of ischemia [70] and protected heart and kidney of hypertensive mice [71]. These studies exemplified that Y-RNA fragments are loaded into exosomes when they are overexpressed in the producer cell. Corroborating with these observations is another study where Influenza A virus infected cells were shown to overexpress Y5-RNA derived small RNA hsa-miR-1975. MiR-1975 was also found to be enriched in exosomes and induced interferon expression from neighboring cells spreading innate anti-viral effect [72]. Collectively, these studies establish Y-RNA as part of metabolic overload which are packaged into exosome for cellular communication and signaling. More research is required to understand the mechanism of redirection of Y-RNA from the producer cell nucleus into exosomes and how this impacts cellular communication and gene expression. 

## 7. LncRNA

Long non-coding RNAs constitute ~3% of total exosomal RNA content derived from human plasma [49]. Several low expressing lncRNAs, p21, CCND1, and HOTAIR have been found to be enriched in exosomes from HeLa cells indicating selective packaging of these lncRNAs [73]. The lncRNA PRINS was found to be a diagnostic indicator in a study comparing multiple myeloma patients to healthy individuals with a sensitivity of more than eighty percent [74]. Exosome associated lncRNAs have been shown to interact with RNA binding proteins of recipient cells L-lactate dehydrogenase B, CSF2Rb and high-mobility group protein-17 to enhance cell viability of recipient cells [75]. 

The therapeutic potential of exosome associated lncRNAs has been realized in several studies. MSC derived exosomal delivery of lncRNA NEAT was found to enhance recovery of cardiac function in vivo compromised by the drug doxorubicin [76]. Exosome packaged lncRNA UCA1 was shown to function in two differing roles, highlighting the duplicity of many lncRNAs in mechanistic function [77,78]. MSC derived exosomes containing UCA1 have been shown to sponge miR-873 and prevent apoptosis to ultimately aid in cardio-protection in myocardial infarction model of rat. UCA1 is also found elevated in serum derived exosomes from patients surviving myocardial infarction, suggesting additional clues towards its protective role [77]. However, UCA1 is not always protective. UCA1 packaged into exosomes derived from cancer associated fibroblasts has been found to sponge miRNA-103 in vulvar squamous cell carcinoma, resulting in enhanced cell proliferation and drug resistance in vitro and in mouse models [78]. Although two different outcomes physiologically, exosomal lncRNA UCA1 has anti-apoptotic effects at the cellular level in recipient cells. Another exosome associated lncRNA is H19. H19 is enriched in MSC exosomes and found to inhibit miR-let7b in exosomes receiving trophoblast cell lines like HTR-8/SV neo cells and prevents their apoptosis [79].

In all these studies lncRNAs are shown to be enriched in exosomes endogenously. However, to realize the full potential of lncRNA in the context of gene regulation, a means to specifically enrich lncRNAs into exosomes will be required. Towards this goal, several investigators have attempted to understand selective export of lncRNAs into exosomes. Specific U6 promoter driven lncRNA overexpression was found to enrich exosomes with transcripts [80]. These authors predicted that there may be a lncRNA specific exosome packaging signal, but did not demonstrate the activity of the discussed sequence per se. Exosomal lncRNAs and mRNAs have been found to possess one or all of three structural motifs; (A) ACCAGCCU, (B) CAGUGAGC and (C) UAAUCCCA, discovered using a bioinformatics based approach [81]. Multifunctional protein YB1 and methyltransferase NSUN, which are enriched in HEK293T derived exosomes, have been found to interact with these motifs, suggesting that there may be a protein-lncRNA exosome packaging complex involved in the packaging of lncRNAs into exosomes [82]. However, this complex is not clear and how this purported exosome packaging complex would mechanistically affect endogenous lncRNA function or to how, or if, this proposed protein complex dissociates from the lncRNA in the target cell to impart function remains to be determined. 

## 8. Circular RNA

Exosome associated circular RNAs were studied in detail in [83] where more than ~1200 different circRNAs were found in human plasma derived exosomes, including several biomarker candidates and many involved in cancer. CircRNA CDYL was found to be enriched in exosomes in the serum of human cancer cell MHCC-LM3 implanted nude mice. The circRNA KLDHC10 was observed in serum exosomes of colorectal cancer patients [83]. The circRNA NRIP has been found to be upregulated in plasma form gastric cancer patients and NRIP enriched in exosomes from gastric cancer cells have found to promote tumor metastasis in vivo [84]. Notably, this study provided the first proof of its kind where circRNA content of exosomes has been found to have a distal effect. 

The utility of exosome associated circRNAs as biomarkers has been shown in studies with circRNA KIAA1244, which has been shown to be over-expressed in gastric cancer patients plasma derived exosomes [85]. Similarly, plasma exosome bound hsa_circ0001946 expression levels have been used to predict the reoccurrence, overall survival and disease-free survival in esophageal squamous cell cancer patients [86]. The circRNA, hsa_circ0004771 has been shown to function as a novel diagnostic biomarker of CRC [87], while adipocyte-derived exosomal hsa_circ0075932 promoted cell inflammation, apoptosis and delayed wound healing of burned skin in obese persons [88]. Exosome associated circRNAs have also been shown to affect paracrine gene expression in diabetes related stress retinal vascular dysfunction mouse models [89]. These are just a few of the many examples of exosome associated circRNA function, clearly indicating that there must be some molecular pathways involved in the packaging and/or recruitment of circRNAs to exosomes.

Studies into the packaging of circRNAs into exosomes are rooted in bioinformatics. Extensive bioinformatics analysis on exosomal specific circRNA sequencing data indicate 5′-GMWGVWGRAG-3′ purine rich motif which is involved in selective packaging of circRNAs with help of particular RNA binding proteins [90]. Several circRNAs are known to be elevated with cancerous cells and are also present in the exosomes derived from the cancerous cells. However, it is not clear whether cells dump circRNAs into exosomes to lower the intracellular concentration of circRNAs, as circRNAs are not believed to be degraded but rather diluted through cell division, or to perhaps promote cancer cell growth into a new niche. Alternatively, similar to lncRNAs, the exosome associated circRNAs may exhibit a multitude of varying functions in target cells that remain to be determined.

## 9. mRNA

Messenger RNA are generally some of the largest transcripts packaged into exosomes. For decades functional mRNAs have been observed to be present in exosomes [91]. Exosome mRNA content is however not proportional to the source cellular RNA content. Approximately 15% of cellular mRNAs were found to be detected in exosomes derived from mast cell line HMC-1 but differed in profile from the parent cellular population [92]. Copies of mRNA per exosome were shown to range from one mRNA per thousand to one mRNA per hundred thousand exosomes in glioma cells [93]. Exosomal mRNA content was also found to be sensitive to oxidative stress [94]. Promyelocytic leukemia cells (NB4) have been observed to secrete exosomes rich in mRNA for the oncogenic fusion protein PML-RARα. Notably, the exosome mRNA expression of PML-RARα enhanced survival of those endothelial cells receiving exosomes in vitro thereby transferring angiogenic signature [95]. While we have listed only a few examples of exosome mRNAs, it is clear that exosomes are used by the cell to disseminate mRNAs to target cells and that these mRNAs are functionally expressed. 

Although cellular and exosomal mRNA populations have been shown to be variable, some researchers have attempted cellular overexpression of desired mRNAs in an attempt to packaging mRNA in exosomes, banking on the garbage theory of exosomes. The over-expression of mRNAs has resulted in notable enrichment of the particular mRNAs in exosomes. One study demonstrated that overexpression of connective tissue growth factor CCN2 in hepatic cells led to CCN2 mRNA enrichment in exosomes which were able to express CCN2 in recipient cells [96]. Overexpression of cystic fibrosis transmembrane conductance regulator (CFTR) vector in A549 cells has also led to the packaging of CFTR protein and mRNA into exosomes. These exosomes restored CFTR defect in CF cells [97]. In a recent study, overexpression was coupled with cellular nanoporation using electrical pulse to drastically boost exosome production and enhance mRNA transcript packaging inside exosomes. With this methodology researchers achieved ~3 log fold increase in PTEN mRNA within exosomes and these exosomes were shown to be effective against PTEN deficient glioma mouse model [98]. 

Another means to enhance mRNA packaging into exosomes could be via the exploitation of RNA protein interactions that encourage loading therapeutic or reporter mRNAs into exosomes. Cytoplasmic protein YB1 has been observed in the exosomal extract from several different cell types, and is known to interact with degenerate consensus sequences on 3′UTR of mRNA to facilitate their exosomal packaging [99]. Whether introduction of these sequences can enhance packaging of desired mRNAs however remains to be elucidated. A similar approach, whereby the exosome marker protein CD63 was fused to archaebacterial RNA interacting protein L7ae, has demonstrated functional packaging and spread of a transgene mRNA [100]. The CD63-L7ae fusion protein facilitated the packaging of mRNA containing the L7ae C/D box recognition sequence in the 3′UTR. This strategy not only enriched mRNA into exosomes but also yielded functional proteins in recipient cells [100]. Capsid forming proteins like Drosophila Arc1 and Copia retrotransposon Gag1 protein have also been shown to interact with consensus RNA sequences and load them into exosomes by interfering with the ESCORT pathway [101]. These approaches are evolving ultimately towards a means to preferentially package mRNAs, in a manner that allows the functional transmission of a therapeutic transgene by exosomes to target cells. Such an approach will not only serve as a revolutionary technology to control gene expression but could also be applied in vivo in a relatively straightforward manner.

## 10. DNA

Exosomes from various sources were found to carry fragments of genomic, mitochondrial and viral DNA [102,103,104]. Exosomes derived from normal neural stem cells were compared to a neuroblastoma cell line or glioblastoma cells, were shown to package DNA harboring mutations in the form of SNPs in the SOX2 gene or insertions in the NANOGP8 gene [105,106] These differences can be harnessed to establish exosome based diagnostic biomarkers for hard-to-reach cancers. Interestingly, these fragments of DNA can act as donor DNA following double stranded break repair in the recipient cell. Supporting this notion are observations of intra-species horizontal gene transfer, which was observed in exosomes containing bovine DNA, from fetal bovine serum added as supplement to DMEM used for culturing NIH-3T3 cells [107]. These bovine DNA containing exosomes were found to become inserted into CRISPR-induced double stranded breaks, suggesting that exosome transfer of DNA can have meaningful stable impacts on recipient cells [107]. Exosome associated DNA can in some cases also lead to consequential protein synthesis in recipient cells. For example, exosome mediated transfer of hybrid gene BCR/ABL between K562 cells to normal neutrophils or in vivo resulted in the expressed protein contributing to pathophysiology of chronic myeloid leukemia [108,109]. In addition to fragments of genomic DNA, mtDNA was also detected in exosomes [110,111]. Fragments of mtDNA were observed to be present in exosomes which can be assembled into whole mitochondrial genome in recipient cells. Two studies completely different in their methodology and outcome have a unifying underlying principal that transfer of mtDNA fragments via exosome led to increase in mtDNA in recipient cells. This resulting increase in copies of mitochondrial DNA in recipient cells results in increased cellular respiration and help recipient cells escape metabolic quiescence [110,111]. Collectively, exosome associated DNAs appear to be a means to both facilitate neighboring cell metabolic function as well as impart signaling and gene expression changes. Such observations suggest that no cell is alone and that collections of cells are linked together with exosomes forming a significant backbone fabric of these intracellular connections.

## 11. Available Toolkit for Developing Exosome-Based Therapies

Currently the scientific world is divided over how best to leverage exosomes as vehicles for delivery of nucleic acid. Attempts to manipulate exosomes after their harvest results in significant loss of biomaterial and is also limited by adsorption of nucleic acid rather than packaging. So, manipulation of exosome producer cells to package required nucleic acid is a viable alternative. Towards this goal, several different research groups have followed different protocols. We are attempting to represent packaging methodologies via three different models. These models can guide researchers in designing packaging methodologies depending on complexity of nucleic acids and malleability of producer cells.

### 11.1. Cellular Abundance Induced Packaging Model 

When a particular nucleic acid is overexpressed in cell either from a plasmid or due to change in cellular microenvironment, the excess nucleic acid gets routed in MVB and gets exported as exosomal cargo. To exemplify, delivery of exosomal tsRNA-10277 [61], circRNA NRIP [84] or mRNA [98] fit in this model. Induction of double stranded breaks in genomic DNA yields excess DNA fragments which gets packaged in the exosome [106]. This model provides easiest way to adopt exosome as vehicle for therapeutic nucleic acid delivery. However, this abundance induced loading can be hampered by size of nucleic acid. (Figure 1).

### 11.2. Cellular Protein Assisted Packaging Model 

Several RNA and DNA binding proteins have been shown to be localized in exosomes via mass spectrometry. When these proteins are enriched in exosomes, their partner nucleic acid can be expected to be enriched in exosomes as well. Banking on this miR-155 was shown to be enriched in exosomes when its binding partner FMR1 was localized to exosome as result of inflammation [38]. Similarly, YB1 protein can be expected to enrich its lncRNA or mRNA binding partners into exosomes [81,99]. Exosome specific RNA motifs were reported in [112]. Mitochondrial DNA fragments, so abundantly reported to be enriched in exosomes, are also bound to protein transcription factor of mitochondria (TFAM) found to be enriched in exosomes [111]. These observations pave way for researchers to design and express nucleic acid with embedded binding sequence to facilitate their sorting in exosomes. This pathway can facilitate packaging of longer nucleic acid but dependency on endogenous cellular factors could potentially be rate limiting. With the recent availability of database of nucleic acid binding proteins [113] and exosome specific proteins [114], an intersectional subset can reveal more such proteins which can carry cognate nucleic acid. This methodology can further be useful in developing tissue specific therapeutic exosomes as lot of RNA binding proteins are tissue specific in expression. (Figure 2).

### 11.3. Engineered Protein Mediated Packaging Model 

Fusion proteins of known exosomal markers and RNA binding proteins were engineered to sequester required RNA in exosome. Authors Li et al. constructed CD9-HuR for enhanced interaction with miR-155 and its subsequent enrichment in exosomes [115]. Authors Kojima et al. went a step further by expressing fusion protein CD63-L7ae and catalase mRNA with C/D box sequence at 3′UTR. L7ae binds with C/D box sequence so that CD63 can sequester catalase mRNA to exosome [100]. A few more examples based on same principal can be found in literature [116,117]. These engineered models are more robust as they encash on the interaction between selected RNA sequence and its high binding affinity protein partner resulting in one to two log fold increase in packaging of specific RNA. However, these strong interactions can also potentially inhibit downstream functioning of RNA. Additionally, these exogenous fusion proteins could potentially be immunogenic in nature. (Figure 3).

## 12. Conclusions

Several studies discussed here exemplify the notion that nucleic acid, RNA and DNA, exosome associated cargo provides a function in altering gene expression in recipient cells in a paracrine fashion. While drawing from various published works here, the observations to date clearly demonstrate that irrespective of selective packaging or passive loading of over-produced transcripts, cells endogenously package nucleic acids into exosomes. This basic cellular biological activity, of packaging valuable nucleic acids into exosomes and sharing these exosomes with neighboring cells, suggests that cells are programed with an inherent nature to interact with their surroundings in a positive altruistic-like manner. Exosomes are the means by which the cellular city is built, allowing for an efficient means of both recycling cellular products and using endogenous cellular properties to disseminate information. This cellular information is coded in those nucleic acids, RNAs and DNAs, packaged into the cellular exosomes. Indeed, if there is an exosome nucleic acid code involved in intracellular communication one would expect it to be a unique property imbued in the exosome by the exosome producer cell. Supporting this notion are the observations whereby tumor cell exosomes have been shown to promote chemoresistance whereas exosomes from stem cells have demonstrated regenerative potential. These observations not only suggest that exosomes imprint the phenotype of their parent producer cell on the recipient cell, but also that the recipient cells must also contain an ability to respond to the exosome nucleic acid signature. Surely unlocking this secret would be to decode the cell. The results of such an understanding of cellular exosome signaling would not only lead to an unparalleled ability to control cellular function but also provide significant insights into understanding disease and cellular aging.

## Figures and Tables

**Figure 1 genes-12-00173-f001:**
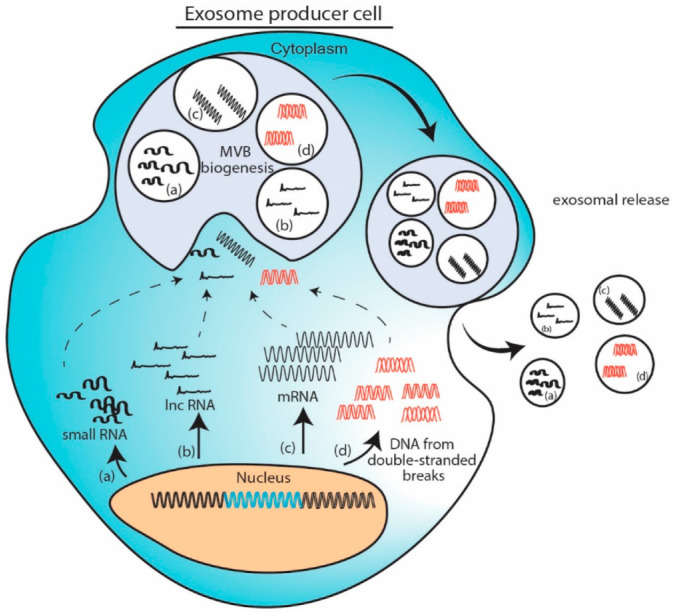
Cellular abundance induced packaging model. Excess nucleic acids in cells like (**a**) small RNA, (**b**) long non-coding RNAs (lncRNA), (**c**) messenger RNA (mRNA) or (**d**) DNA fragments are loaded into exosomes while formation of multivesicular bodies and are exported. Exosomes thus obtained contains overexpressed nucleic acid amongst other cargo.

**Figure 2 genes-12-00173-f002:**
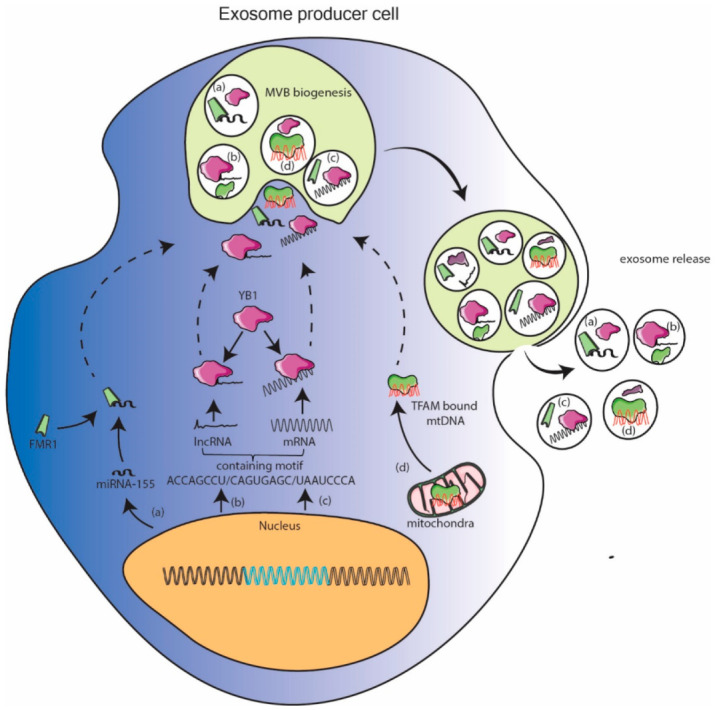
Cellular protein assisted packaging model. Cellular proteins which were found in exosomes can recruit nucleic acids with which they can interact by their inherent nature or with certain sequence. (**a**) enrichment of miR-155 with FMR1, (**b**,**c**) plausible enrichment of certain lncRNA and mRNA with YB1 and (**d**) enrichment of mtDNA with transcription factor of mitochondria (TFAM) are exemplified under this model.

**Figure 3 genes-12-00173-f003:**
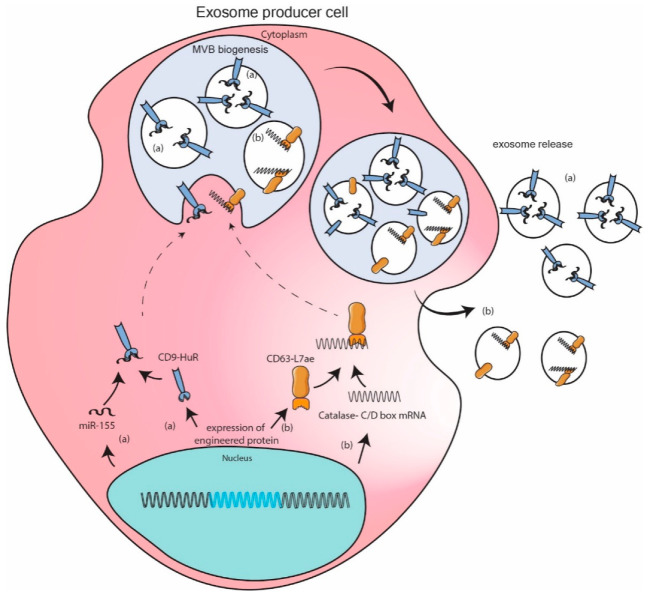
Engineered protein mediated packaging model. Modification of exosome marker protein by fusing them with high affinity RNA binding domain leads to recruitment of specific RNA in exosome. Example depicted here are (**a**) interaction on micro RNA (miR)-155 with HuR fused to exosomal marker protein CD9 enriches miR-155 in exosome. (**b**) catalase mRNA containing C/D box sequence at 3′UTR can bind to L7ae fused to exosomal marker protein CD63 leading to enrichment of catalase mRNA in exosomes.

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
