# Peer review of "The Multifunctionality of Exosomes; from the Garbage Bin of the Cell to a Next Generation Gene and Cellular Therapy"

_genes, 2021, doi:10.3390/genes12020173_

Round 1

Reviewer 1 Report

In this review titled “The multifunctionality of exosomes; from the garbage bin of the cell to a next generation gene and cellular therapy” by Shrivastava and Morris, the authors provide an overview of the various nucleic acid populations present within, or associated with, exosomes and provide examples to argue that they are not merely cellular debris (as initially thought) but mediators crucial for intercellular communication. While the review is on an interesting and important topic, I feel the manuscript can be improved as I describe in detail below.

Major points:

1, First of all, I would like to commend the authors for the numerous examples collated for each nucleic acid population within/associated with exosomes. However, I felt that the review could be better organized. The numerous examples (mostly independent of each other) presented in tandem somewhat distracts the reader from the main point that each paragraph is trying to get across. In this regard, I think it is fairly consensus by now that exosomes are important players in intercellular communication and not merely cellular debris. As such, I think it would greatly improve the manuscript if the authors could further provide a unique contribution to the expanding field especially by overviewing and discussing how exosomes (or its biology) may be leveraged for “next-generation cellular and gene therapy”. A friendly suggestion is to expand and dedicate a separate section to the discussion regarding the current limitations of the field and detailed suggestions for how to overcome them (how can exosomes be best leveraged to be utilized in diagnosis/treatment? What are the challenges that need to be overcome? How can the challenges be overcome?)

2, Also a friendly suggestion: figures/tables will be helpful for readers.

3, Lines 32-33: The authors define EVs as “biological materials surrounded by a lipid bilayer membrane which lack a functional nucleus and vary in size range from 30 to 10000 nm” and cites the consensus statement made by the International Society for Extracellular Vesicles (ISEV) in 2018 (Ref 1). The ISEV2018 statement, to the best of my knowledge, does not mention a size cutoff. The authors should therefore add a reference to support this value. Moreover, I somewhat feel that this value is too large as a red blood cell would also fit the authors’ definition.

4, Lines 35-36: Somewhat related to point #1, the authors provide cutoff values for the various EVs, but the ISEV2018 statement (Ref 1) does not mention such values. I would suggest the authors provide additional references or make it clear that this is the working definition adopted by the authors in this manuscript and perhaps provide a table to ease understanding (as well as explicitly mention other defining features, if any).

5, Line 39: The authors mention “three populations”. I was not entirely sure which three the authors meant, as they seem to have defined five: exomeres, exosomes, microvesicles, oncosomes, and apoptotic bodies—please clarify. I also suggest that the authors at least describe in brief how exosomes (the main subject of this review) are produced.

6, Lines 41-49: The authors mention that exosomes comprise nucleic acids, lipids, and proteins but decide to only discuss nucleic acids. I felt that the rationale for this could be made clearer. In particular, I was not exactly sure what the authors meant by “…which provides an extraordinary opportunity to disseminate proteins (mRNA)” (lines 44-45). If the authors meant to say gene-encoding material, I think it would be more accurate to say so (rather than assume that by the transfer of mRNA, proteins are effectively being disseminated).

7, Line 56: The authors list “circadian rhythm” as a microenvironmental factor alongside cigarette smoke and gamma rays. I think this is misleading, as while the latter two are completely extrinsic, circadian rhythms emerge from an intrinsically wired oscillator that is complexly regulated by the extrinsic world (something like “light exposure” perhaps might be a better term to use?).

8, It would promote the readers’ understanding if the various RNA populations (especially the less well-known ones like piRNA) are briefly explained at the beginning of the respective paragraph (not necessarily in relation to exosomes but in general—what is it? How is it different from the other RNAs?). For some populations, the explanations are already in place, but for others, it is lacking.

9, Lines 365-368: The authors suggest that cells are programmed with “an inherent nature to interact with their surroundings in a positive altruistic-like manner “…based on the fact that cells “packag[e] valuable nucleic acids into exosomes, and shar[e] these exosomes with neighboring cells”. I find this assertion rather problematic as it conflates a biological phenomenon (exosomes as vehicles of cellular communication) with a social one (altruism). This conflation is driven by the authors’ views that nucleic acids are “valuable” and that the exosomes are “shared”, but I do not think this is substantiated by experimental evidence.

Minor point:

1, Line 128: Did the authors mean to say “2’-O-methylation” instead of “2’0 methylation”?

Author Response

We thank the reviewers for their comments regarding this manuscript. Our responses are below in italics and underlined.

Reviewer 1

Major points:

1, First of all, I would like to commend the authors for the numerous examples collated for each nucleic acid population within/associated with exosomes. However, I felt that the review could be better organized. The numerous examples (mostly independent of each other) presented in tandem somewhat distracts the reader from the main point that each paragraph is trying to get across. In this regard, I think it is fairly consensus by now that exosomes are important players in intercellular communication and not merely cellular debris.

Detailed exosome biodistribution studies reveal that systemically majority of exosome gets expelled along with urine and/or routed to liver for degradation as reviewed in (1) and also observed in biodistribution experiments in our lab. Intracellularly as well, a significant portion of uptaken exosomes are diverted to late endosome à lysosome pathway and only a fraction of exosome cargo gets exposed to cytosol (2) (https://www.biorxiv.org/content/10.1101/2020.10.16.341974v2.full) and also observed in our lab. Exosomes are known to be packed with cellular overproduce (many examples cited throughout the manuscript). Additionally packaging RNA in exosomes helps in regulation of half life of RNA (3). So overall it can not be ruled out that for producer cells exosomes serves as a means to dump the garbage. However, intercellular communication appears to be a secondary but important function achieved by exosomes. This review intends to highlight multifaceted nature of exosomes and ways to harness them.

As such, I think it would greatly improve the manuscript if the authors could further provide a unique contribution to the expanding field especially by overviewing and discussing how exosomes (or its biology) may be leveraged for “next-generation cellular and gene therapy”. A friendly suggestion is to expand and dedicate a separate section to the discussion regarding the current limitations of the field and detailed suggestions for how to overcome them (how can exosomes be best leveraged to be utilized in diagnosis/treatment? What are the challenges that need to be overcome? How can the challenges be overcome?)

Response: A separate section on  “available toolkit for developing exosome-based therapies” has been added.

2, Also a friendly suggestion: figures/tables will be helpful for readers.

Response: Figures on proposed model for packaging methodologies has now been added.

3, Lines 32-33: The authors define EVs as “biological materials surrounded by a lipid bilayer membrane which lack a functional nucleus and vary in size range from 30 to 10000 nm” and cites the consensus statement made by the International Society for Extracellular Vesicles (ISEV) in 2018 (Ref 1). The ISEV2018 statement, to the best of my knowledge, does not mention a size cutoff. The authors should therefore add a reference to support this value. Moreover, I somewhat feel that this value is too large as a red blood cell would also fit the authors’ definition.

Response: We apologize for wrong citation listing. Correct citation has now been added. Regarding size of EVs large oncosomes have been reported to be in range of 10,000 nm range and ae classified as EVs (4). Maybe that’s why reports on the occurrence of what we now call EVs were first published in the late 1960s, with researchers referring to observed extracellular structures or lipid-rich particles as “platelet-dust” or “matrix-vesicles” (5). So early researchers also compared them to blood cells and observed that they lacked a nucleus.

4, Lines 35-36: Somewhat related to point #1, the authors provide cutoff values for the various EVs, but the ISEV2018 statement (Ref 1) does not mention such values. I would suggest the authors provide additional references or make it clear that this is the working definition adopted by the authors in this manuscript and perhaps provide a table to ease understanding (as well as explicitly mention other defining features, if any).

Response: We apologize for the wrong citation listing and correct citation is added. However listing characteristics of each kind of EV was done in recent review {31311206} and would be much beyond the scope of current review manuscript focusing of packaging of nucleic acids in exosomes.

5, Line 39: The authors mention “three populations”. I was not entirely sure which three the authors meant, as they seem to have defined five: exomeres, exosomes, microvesicles, oncosomes, and apoptotic bodies—please clarify. I also suggest that the authors at least describe in brief how exosomes (the main subject of this review) are produced.

Response: We apologize for this mistake and we have now added a brief description of exosome biogenesis is added in the introduction section.

6, Lines 41-49: The authors mention that exosomes comprise nucleic acids, lipids, and proteins but decide to only discuss nucleic acids. I felt that the rationale for this could be made clearer. In particular, I was not exactly sure what the authors meant by “…which provides an extraordinary opportunity to disseminate proteins (mRNA)” (lines 44-45). If the authors meant to say gene-encoding material, I think it would be more accurate to say so (rather than assume that by the transfer of mRNA, proteins are effectively being disseminated).

Response: As per reviewer’s recommendation, changes have been incorporated in the manuscript.

7, Line 56: The authors list “circadian rhythm” as a microenvironmental factor alongside cigarette smoke and gamma rays. I think this is misleading, as while the latter two are completely extrinsic, circadian rhythms emerge from an intrinsically wired oscillator that is complexly regulated by the extrinsic world (something like “light exposure” perhaps might be a better term to use?).

Response: Whether extrinsic (cigarette smoke, gamma rays ) or intrinsic (circadian rhythm) , said factors influence the cellular microenvironment which causes change in exosomal miRNA profile. The sentence was meant to highlight the sensitivity and dynamic nature of exosomal miRNAs and not meant to be misleading.

8, It would promote the readers’ understanding if the various RNA populations (especially the less well-known ones like piRNA) are briefly explained at the beginning of the respective paragraph (not necessarily in relation to exosomes but in general—what is it? How is it different from the other RNAs?). For some populations, the explanations are already in place, but for others, it is lacking.

Response: A brief description about piRNAs has now been added to the manuscript as per reviewer’s recommendation.

9, Lines 365-368: The authors suggest that cells are programmed with “an inherent nature to interact with their surroundings in a positive altruistic-like manner “…based on the fact that cells “packag[e] valuable nucleic acids into exosomes, and shar[e] these exosomes with neighboring cells”. I find this assertion rather problematic as it conflates a biological phenomenon (exosomes as vehicles of cellular communication) with a social one (altruism). This conflation is driven by the authors’ views that nucleic acids are “valuable” and that the exosomes are “shared”, but I do not think this is substantiated by experimental evidence.

Response: We were under the premise that the conclusion section is for authors speculations based on the reviewed literature. Degradation of RNA is endothermic process for cell so sharing RNAs via exosomes can prove to be positively altruistic and thus evolutionarily conserved trait. Experimental evidence of sharing nucleic acids via exosomes are in accordance of their ability to act as inter cellular communicators as exemplified by many citations throughout the manuscript.

Minor point:

1, Line 128: Did the authors mean to say “2’-O-methylation” instead of “2’0 methylation”?

Response: We apologize for the typographical error and correction has been made in manuscript.

Reviewer 2 Report

Shrivastava et al reviewed the cargos in exosome/extracellular vesicles (EVs) that may contribute to the regulation of cellular properties. Be specific, the authors reviewed and categorized cargos into small RNAs, lncRNAs, mRNAs and DNAs in the exosome/EVs. This manuscript is well written, and literature and examples are well aligned with the topic. However, most of the literature and reference is listed "as is" without any comments, correlations, analysis from the authors, especially the sorting and packaging mechanisms of these different cargos into exosomes/EVs. This part needs more input from the author to further improve the quality of this manuscript. Otherwise, I feel the topic and review is critical for the broader audience interested in exosome field.

1, line 39-40, “out of three populations”, not sure which three populations the authors are referring to?

2, line 76-77, please expand the analysis of this reference, any reason for capitalizing the “H” in human PTEN?

3, the authors did a good job on categorizing the cargos (RNAs and DNAs) in the exosomes and also provided examples and literatures for their discovery and potential functions. However, the beacon of this review should be finding the similarity/difference in the sorting/packaging mechanisms of these cargos, thus providing information on the applications of exosomal RNA/DNA in diagnosis or therapy for different cell type or disease type. I barely see any discussions in the sorting mechanisms. For instance, the most-widely studied miRNAs in exosome/EVs have different sorting mechanisms. Please add this analysis/discussion to each section of the manuscript instead of listing references and examples.

4, I am really curious about some the RNAs that the author mentioned, e.g. snoRNAs, piRNAs and yRNAs. Are they widely presented in all cell-derived EVs/exosomes? Or they are enriched in certain populations of EVs? Or they are cell-type specific? Or even disease specific? I would like to see the comments/discussions from the authors.

5, line 235-237, the font is completely off.

6, how the exosomal RNAs and DNAs discussed in this review are associated with cellular aging? As the authors concluded.

Author Response

Reviewer 2

Shrivastava et al reviewed the cargos in exosome/extracellular vesicles (EVs) that may contribute to the regulation of cellular properties. Be specific, the authors reviewed and categorized cargos into small RNAs, lncRNAs, mRNAs and DNAs in the exosome/EVs. This manuscript is well written, and literature and examples are well aligned with the topic. However, most of the literature and reference is listed "as is" without any comments, correlations, analysis from the authors, especially the sorting and packaging mechanisms of these different cargos into exosomes/EVs. This part needs more input from the author to further improve the quality of this manuscript. Otherwise, I feel the topic and review is critical for the broader audience interested in exosome field.

1, line 39-40, “out of three populations”, not sure which three populations the authors are referring to?

Response: We apologize for the mistake and changes have been incorporated in the manuscript.

2, line 76-77, please expand the analysis of this reference, any reason for capitalizing the “H” in human PTEN?

Response: As per reviewer’s recommendation, changes have been incorporated in the manuscript.

3, the authors did a good job on categorizing the cargos (RNAs and DNAs) in the exosomes and also provided examples and literatures for their discovery and potential functions. However, the beacon of this review should be finding the similarity/difference in the sorting/packaging mechanisms of these cargos, thus providing information on the applications of exosomal RNA/DNA in diagnosis or therapy for different cell type or disease type. I barely see any discussions in the sorting mechanisms. For instance, the most-widely studied miRNAs in exosome/EVs have different sorting mechanisms. Please add this analysis/discussion to each section of the manuscript instead of listing references and examples.

Response: As per reviewer’s recommendation a separate section on “available toolkit for developing exosome-based therapies” has now been added. Also figures on proposed model for packaging methodologies have been added.

4, I am really curious about some the RNAs that the author mentioned, e.g. snoRNAs, piRNAs and yRNAs. Are they widely presented in all cell-derived EVs/exosomes? Or they are enriched in certain populations of EVs? Or they are cell-type specific? Or even disease specific? I would like to see the comments/discussions from the authors.

Response: The reviewers curiosity is highly appreciated. However, in absence of database of snoRNAs, piRNAs and Y-RNA in exosomes from various cell types, it cannot be confidently commented that they are present in exosomes from all types of sources. Plasma derived exosomes, which are assumed to be representative of exosomes from a large variety of cells, have shown presence of these unique RNAs which is cited in manuscript in their respective sections.

5, line 235-237, the font is completely off.

Response: We apologize for the mistake and changes have been incorporated in the manuscript.

Reviewer 3 Report

In their manuscript titled "The multi functionality of exosomes; from the garbage bin of the cell to a next generation gene and cellular therapy" Shrivastava and Morris provide a brief review of the very wide array of studies exploring the biology of nucleic acids packaged in exosomes.

The article is well written and organized. I would imagine that such would be the impact on a broader audience. On the other hand, for someone who might be interested in digging deeper into the field, I would imagine that they would find this to be a useful review for it shares a very thorough survey of the many articles published on the subject.

However, I would also hope that the authors carefully consider some of my recommendations below, all of which are made with the intention of improving the manuscript even further.

MAJOR EDITS

1) The manuscript would greatly benefit from a brief review of how EVs (and in particular, exosomes) are formed in source cells. What are some of the known molecular pathways and mechanisms involved in their production? While the authors provide references to articles that describe this in detail, and while reviewing the mechanisms of exosome production is not the main focus of this manuscript, the general readership would nonetheless greatly benefit from a brief description of how exosomes are formed.

2) In L118, the authors indicate that 2'O methylation or acetylation on an snoRNA's complementary RNA leads to "tightening the gene expression regulation". This requires further explanation. What do the authors mean by "tightening gene expression regulation"? This is likely a mechanism that is unknown for a broader audience, so the authors may need to add 1-2 sentences to further elaborate on this concept.

3) In L237 the authors mention YB1 and NSUN as proteins that mediate the loading of exosomes of sequences carrying specific motifs. As a general audience reader, I found myself wondering: do we know anything about the epidemiological correlation between these proteins and disease? Aside from the work cited, do we know if there exists any type of correlation between dysfunction in these proteins and cardiac protection or prognosis in vulvar squamous cell carcinoma?

4) L332-335. It is unclear what the authors mean by neural stem cells "relative" to a neuroblastoma cell line or glioblastoma cells. This phrase needs revising.

5) L343-344. "Exosome associated ... recipient cells". What do the authors mean by "adherent protein synthesis"? What adherent proteins? This phrase lacks context and needs revising.

6) L366-368: "...cells are programed with an inherent nature to interact with their surroundings in a positive altruistic-like manner..." This phrase assumes that there is an adaptive value to intercellular communication through EV. If that were the case, you'd expect that cells/organisms that can't produce exosomes loose adaptive potential. Is there any evidence of that? Have any studies inhibited the capacity of cells or organisms to produce exosomes and shown a detriment for target cells/organs? Because, an alternative possibility is that EV formation evolved for its adaptive value from the producing cell (e.g. to discard excess material), and that effects on target cells represent only secondary consequences.

MINOR EDITS/CORRECTIONS

1) The manuscript needs careful proofreading for minor grammatical mistakes spread throughout the article, most often related to missing articles or pronouns. The list below is just a brief subset of some to illustrate my point. There are other minor fixes needed throughout the manuscript that are not included here - please proofread carefully:

L35, "nm" missing in (50 to 150)
L42, "Nucleic acid, [INCLUDING] messenger..."
L44, delete "which" - "provides" should be "provide"
L72, "s" missing in vessel
L73, "...stenosis in AN atherothrombosis ..."
L156 "...patients HAVE (not has) been found..."

2) Some sentences need careful revision for structure - the idea that the authors wanted to communicate is unclear . For instance: L76-78; L79-81; L127-129

3) In L155, does "EV" stand for extracellular vesicle? Probably - but it needs to be clarified just in case.

4) L235 & L237 - Is there a problem with the font size in "structural motifs" and "bionformatic based"? And by the way, it should probably be "bioinformaticS based approach".

5) L363: "...observations to date do not rule out..." - In this phrase, "do not rule out" sounds too cautious and passive. The whole manuscript is meant to emphasize the importance of the packaging of nucleic acids into exosomes, regardless of mechanism. In this regard, a more assertive term, such as "clearly indicate" or "demonstrate" is probably more appropriate.

6) L380: "...ability TO control..."

Author Response

Reviewer 3

Suggestions for Authors

In their manuscript titled "The multi functionality of exosomes; from the garbage bin of the cell to a next generation gene and cellular therapy" Shrivastava and Morris provide a brief review of the very wide array of studies exploring the biology of nucleic acids packaged in exosomes.

The article is well written and organized. I would imagine that such would be the impact on a broader audience. On the other hand, for someone who might be interested in digging deeper into the field, I would imagine that they would find this to be a useful review for it shares a very thorough survey of the many articles published on the subject.

However, I would also hope that the authors carefully consider some of my recommendations below, all of which are made with the intention of improving the manuscript even further.

MAJOR EDITS

1) The manuscript would greatly benefit from a brief review of how EVs (and in particular, exosomes) are formed in source cells. What are some of the known molecular pathways and mechanisms involved in their production? While the authors provide references to articles that describe this in detail, and while reviewing the mechanisms of exosome production is not the main focus of this manuscript, the general readership would nonetheless greatly benefit from a brief description of how exosomes are formed.

Response: The reviewer has rightly identified that biogenesis of EVs is not the major focus of this manuscript focused on nucleic acid packaging in exosomes. A brief description of exosome biogenesis has now been added in the introduction per reviewer’s suggestion.

2) In L118, the authors indicate that 2'O methylation or acetylation on an snoRNA's complementary RNA leads to "tightening the gene expression regulation". This requires further explanation. What do the authors mean by "tightening gene expression regulation"? This is likely a mechanism that is unknown for a broader audience, so the authors may need to add 1-2 sentences to further elaborate on this concept.

Response: The suggested changes have now been incorporated into the manuscript as per the reviewers suggestion.

3) In L237 the authors mention YB1 and NSUN as proteins that mediate the loading of exosomes of sequences carrying specific motifs. As a general audience reader, I found myself wondering: do we know anything about the epidemiological correlation between these proteins and disease? Aside from the work cited, do we know if there exists any type of correlation between dysfunction in these proteins and cardiac protection or prognosis in vulvar squamous cell carcinoma?

Response: Roles of YB1 are discussed in reviews (6, 7), NSUN (8, 9) Discussing role of each exosomal protein and their epidemiological correlation between disease would be beyond the scope of this review focused on nucleic acid packaging in exosomes.

4) L332-335. It is unclear what the authors mean by neural stem cells "relative" to a neuroblastoma cell line or glioblastoma cells. This phrase needs revising.

Response: I apologize for the mistake and changes have been incorporated in the manuscript.

5) L343-344. "Exosome associated ... recipient cells". What do the authors mean by "adherent protein synthesis"? What adherent proteins? This phrase lacks context and needs revising.

Response: Adherent protein synthesis was meant for consequent protein systhesis following RNA delivery via exosome. As per reviewer’s recommendation, changes have been incorporated in the manuscript.

6) L366-368: "...cells are programed with an inherent nature to interact with their surroundings in a positive altruistic-like manner..." This phrase assumes that there is an adaptive value to intercellular communication through EV. If that were the case, you'd expect that cells/organisms that can't produce exosomes loose adaptive potential. Is there any evidence of that? Have any studies inhibited the capacity of cells or organisms to produce exosomes and shown a detriment for target cells/organs? Because, an alternative possibility is that EV formation evolved for its adaptive value from the producing cell (e.g. to discard excess material), and that effects on target cells represent only secondary consequences.

Response: EV’s have been shown to be produced from all organisms from prokaryotes (10), eukaryotes including protists (11), plants (12) and animals. So, organisms not producing EV are not known. Inhibition of exosome secretion is shown to aggravate malfunctioning of cellular metabolism (13). Further, Tipifarnib: an anticancer drug was shown to induce apoptosis at micromolar concentration and a potent inhibitor of exosome production in nanomolar range.(14) In yet another study blocking of exosome release was found to lead cell into cell cycle arrest (15). These studies indicate that excessive block of exosome release may lead to apoptosis. Discussing this further would be beyond the scope of this review. 

MINOR EDITS/CORRECTIONS

1) The manuscript needs careful proofreading for minor grammatical mistakes spread throughout the article, most often related to missing articles or pronouns. The list below is just a brief subset of some to illustrate my point. There are other minor fixes needed throughout the manuscript that are not included here - please proofread carefully:

L35, "nm" missing in (50 to 150)
L42, "Nucleic acid, [INCLUDING] messenger..."
L44, delete "which" - "provides" should be "provide"
L72, "s" missing in vessel
L73, "...stenosis in AN atherothrombosis ..."
L156 "...patients HAVE (not has) been found..."

2) Some sentences need careful revision for structure - the idea that the authors wanted to communicate is unclear . For instance: L76-78; L79-81; L127-129

3) In L155, does "EV" stand for extracellular vesicle? Probably - but it needs to be clarified just in case.

4) L235 & L237 - Is there a problem with the font size in "structural motifs" and "bionformatic based"? And by the way, it should probably be "bioinformaticS based approach".

5) L363: "...observations to date do not rule out..." - In this phrase, "do not rule out" sounds too cautious and passive. The whole manuscript is meant to emphasize the importance of the packaging of nucleic acids into exosomes, regardless of mechanism. In this regard, a more assertive term, such as "clearly indicate" or "demonstrate" is probably more appropriate.

6) L380: "...ability TO control..."

Response: For points 1 to 6 As per reviewer’s recommendation, changes have been incorporated in the manuscript

Literature cited

  1. A. Pinheiro et al., Extracellular vesicles: intelligent delivery strategies for therapeutic applications. J Control Release 289, 56-69 (2018).
  2. B. S. Joshi, M. A. de Beer, B. N. G. Giepmans, I. S. Zuhorn, Endocytosis of Extracellular Vesicles and Release of Their Cargo from Endosomes. ACS Nano 14, 4444-4455 (2020).
  3. A. O. Batagov, V. A. Kuznetsov, I. V. Kurochkin, Identification of nucleotide patterns enriched in secreted RNAs as putative cis-acting elements targeting them to exosome nano-vesicles. BMC Genomics 12 Suppl 3, S18 (2011).
  4. B. Meehan, J. Rak, D. Di Vizio, Oncosomes - large and small: what are they, where they came from? J Extracell Vesicles 5, 33109 (2016).
  5. P. Wolf, The nature and significance of platelet products in human plasma. Br J Haematol 13, 269-288 (1967).
  6. K. Kohno, H. Izumi, T. Uchiumi, M. Ashizuka, M. Kuwano, The pleiotropic functions of the Y-box-binding protein, YB-1. Bioessays 25, 691-698 (2003).
  7. M. Kuwano et al., The basic and clinical implications of ABC transporters, Y-box-binding protein-1 (YB-1) and angiogenesis-related factors in human malignancies. Cancer Sci 94, 9-14 (2003).
  8. M. Wnuk, P. Slipek, M. Dziedzic, A. Lewinska, The Roles of Host 5-Methylcytosine RNA Methyltransferases during Viral Infections. Int J Mol Sci 21, (2020).
  9. K. E. Bohnsack, C. Hobartner, M. T. Bohnsack, Eukaryotic 5-methylcytosine (m(5)C) RNA Methyltransferases: Mechanisms, Cellular Functions, and Links to Disease. Genes (Basel) 10, (2019).
  10. A. Chronopoulos, R. Kalluri, Emerging role of bacterial extracellular vesicles in cancer. Oncogene 39, 6951-6960 (2020).
  11. M. Sharma et al., Characterization of Extracellular Vesicles from Entamoeba histolytica Identifies Roles in Intercellular Communication That Regulates Parasite Growth and Development. Infect Immun 88, (2020).
  12. D. G. Chukhchin, K. Bolotova, I. Sinelnikov, D. Churilov, E. Novozhilov, Exosomes in the phloem and xylem of woody plants. Planta 251, 12 (2019).
  13. Z. He et al., Exosomal secretion may be a self-protective mechanism of its source cells under environmental stress: A study on human bronchial epithelial cells treated with hydroquinone. J Appl Toxicol 41, 265-275 (2021).
  14. A. Datta et al., High-throughput screening identified selective inhibitors of exosome biogenesis and secretion: A drug repurposing strategy for advanced cancer. Sci Rep 8, 8161 (2018).
  15. M. B. Huang, R. R. Gonzalez, J. Lillard, V. C. Bond, Secretion modification region-derived peptide blocks exosome release and mediates cell cycle arrest in breast cancer cells. Oncotarget 8, 11302-11315 (2017).

Round 2

Reviewer 1 Report

Some of the points raised in my previous review were sufficiently improved, and I especially felt the addition of the “toolkit” section greatly adds to the review. However, points #1, #4, and #7 of my previous review report remain inadequately addressed.

#1: As reviewer #2 also mentioned in the first round of review, the examples listed are extensive but mostly without any comments/correlations/analysis from the authors.

#4: It would be great if the authors specifically direct readers to a dedicated review in the text.

#7: I understand the authors’ intentions, but the point remains unaddressed in the manuscript.

Author Response

Reviewer 1

Some of the points raised in my previous review were sufficiently improved, and I especially felt the addition of the “toolkit” section greatly adds to the review. However, points #1, #4, and #7 of my previous review report remain inadequately addressed.

We thank Reviewer 1 for the critical comments and helpful suggestions. We have taken all these comments and suggestions into account, and they have improved our manuscript considerably.

#1: As reviewer #2 also mentioned in the first round of review, the examples listed are extensive but mostly without any comments/correlations/analysis from the authors.

As per the reviewers’ suggestion, the whole section on “Available toolkit for developing exosome-based therapies” was added to comment on variability of methodology of nucleic acid packaging/ draw correlations amongst several different pathways of nucleic acid import in multi-vesicular bodies/ analyze pros and cons of each proposed methodological pathway. Throughout rest of the manuscript, examples are used to justify the opening sentences in each paragraph. We hope that the reviewer #1 finds the section acceptable as reviewer #2 did.

#4: It would be great if the authors specifically direct readers to a dedicated review in the text.

As per the reviewer’s insightful suggestion, word review is added along citation number 11 (line no 48) and 29 (line no 81) in the manuscript.

#7: I understand the authors’ intentions, but the point remains unaddressed in the manuscript.

We thank the reviewer for understanding our intention in this sentence. The objective of the said sentence (line no 72-73) was to mention some examples of factors influencing exosomal miRNA dynamics; our objective was not to classify endogenous from exogenous factors. For this reason, we do not find the sentence to be misleading and thus no change was made in manuscript. We hope the reviewer considers our explanation.

Reviewer 2 Report

The authors have addressed all my concerns. Thanks!

Author Response

(The authors gave the same response as above.)
